# Has the Prevalence of Childhood Obesity in Spain Plateaued? A Systematic Review and Meta-Analysis

**DOI:** 10.3390/ijerph19095240

**Published:** 2022-04-26

**Authors:** Diana M. Bravo-Saquicela, Angelo Sabag, Leandro F. M. Rezende, Juan Pablo Rey-Lopez

**Affiliations:** 1Faculty of Medical Sciences, University Laica Eloy Alfaro de Manabí (ULEAM), Manta 130217, Ecuador; dianabravo_md@outlook.es; 2Faculty of Health Sciences, Valencian International University (VIU), 46002 Valencia, Spain; 3NICM Health Research Institute, Western Sydney University, Westmead, NSW 2145, Australia; a.sabag@westernsydney.edu.au; 4Department of Preventive Medicine, Escola Paulista de Medicina, Universidade Federal de Sao Paulo, Sao Paulo 04023-900, Brazil; leandro.rezende@unifesp.br; 5Faculty of Sport, Catholic University San Antonio of Murcia (UCAM), 30107 Murcia, Spain

**Keywords:** obesity, prevalence, children, population-based study

## Abstract

The prevalence of excess body weight (overweight plus obesity) in children has risen during the last decades in many countries, but it is unclear whether it has reached a plateau in Spanish children. We performed an updated systematic review and meta-analysis for the prevalence of excess body weight in children from Spain, comparing the trends between 1999 and 2010 and 2011 and 2021. Data were reported in a prior meta-analysis, plus an updated search using the Web of Science, MEDLINE (via PubMed) and EMBASE databases for data from January 2018 until December 2021. Thirteen representative studies were identified (34,813 children aged 2 to 13 years), with sample sizes averaging 2678 (range: 396–16,665). The prevalence of excess body weight in Spanish children aged 2 to 6 years increased from 23.3% (95% CI, 18.5% to 25.5%) during the period 1999–2010 to 39.9% (95% CI, 35.4% to 44.7%) during 2011–2021. In children aged 7 to 13 years, the prevalence of excess body weight increased from 32.3% (95% CI, 29.1–35.6%) during the period 1999–2010 to 35.3% (95% CI, 32.9–37.7%) during 2011–2021. The prevalence of childhood overweight and obesity in Spain has substantially increased in the last decade. New food policies to address the childhood obesity epidemic are urgently required to reverse current trends.

## 1. Introduction

The prevalence of overweight and obesity in children rose globally from 1975 to 2016 [1]. Recently, a few nations have introduced (regulatory-based) obesity policies to lower the consumption of ultra-processed food and beverages [2]; however, no nation to date has reversed its obesity epidemic, which continues to stress public health systems [3]. In middle- and low-income countries, it is expected that the prevalence of overweight and obesity among children and adolescents will continue to rise in the coming years [2]. However, amongst high income countries, recent reports suggested a trend for the plateauing or even decreasing prevalence of excess body weight (overweight plus obesity) in children and adolescents [1,4]. For example, between 1975 and 2016, data from the Non-communicable Disease Risk Factor Collaboration (NCD-RisC) showed a flattening in children’s and adolescents’ age-standardized mean BMI in northwestern Europe (both sexes), eastern Europe (girls) and southwestern Europe (boys) [1].

A recent systematic review and meta-analysis involving children (2–13 years old) from 28 European countries found a general stabilization in the rates of obesity across Europe from 1999 to 2016 [4]. In other words, all European countries examined could have reached a simultaneous ceiling in the rates of excess body weight prevalence, backing the existing national food policies to counteract the obesity epidemic. In Spain, the stabilization of obesity was disseminated in the third most read newspaper in the country (elMundo) [5], as a hopeful achievement for public health. [6] Nonetheless, to describe the prevalence of excess body weight (overweight plus obesity) in Spanish children, the previous review [4] included (from a total of 15 studies) 4 non-representative studies. This decision was unfounded as the authors expressed in a published protocol that non-population-based studies would be excluded during the eligibility stage [7]. According to Miguel Porta, population-based is defined as “pertaining to a general population defined by geopolitical boundaries; this population is the denominator and/or the sampling frame” [8]. The recruitment of a convenience sample must not be conflated with a representative sample of the general population of Spanish children (at the national or regional or municipal levels). 

Therefore, we aimed to perform an updated systematic review and meta-analysis of the prevalence of obesity and excess body weight in children from Spain, comparing the trends in two periods, 1999–2010 and 2011–2021. The description of the prevalence of childhood obesity and the prevalence of excess body weight (overweight plus obesity) using updated sources of information is a key goal to evaluate the impact of existing policies employed to tackle the obesity epidemic in this Mediterranean country.

## 2. Materials and Methods 

The systematic review was performed based on the Preferred Reporting Items for Systematic Reviews and Meta-analyses (PRISMA) 2020 guidelines for updated meta-analysis [9] and on the Meta-analysis of Observational Studies in Epidemiology (MOOSE) guidelines [10] for reporting (Appendix A, Appendix A). 

### 2.1. Identification of Previous Studies

Fifteen studies [11,12,13,14,15,16,17,18,19,20,21,22,23,24,25] of prevalence of obesity and excess body weight (overweight plus obesity) identified in a previous systematic review [4] in Spain (from inception until May 2018) were added to the eligibility stage of the systematic review. 

### 2.2. Search Strategy

We searched studies using Web of Science, MEDLINE (via PubMed) and EMBASE databases for studies conducted from January 2018 until December 2021. The search strategy included the following terms: (1) population (children, childhood, schooler, schoolchildren, preadolescent, adolescent, school aged, school-aged); (2) outcome (obesity, overweight, body composition, body constitution, weight status, anthropometry); (3) study design (prevalence, trend, epidemiology, observational, cross-sectional, longitudinal); and (4) location (Spain) (Appendix A). Additionally, we read the reference lists of the included studies in order to identify additional studies. Two authors contributed independently at each stage. The literature search was performed by A.S. and J.P.R.L. and the data extraction by D.M.B.S. and J.P.R.L., and in case of disagreements, a third author made a final decision. 

### 2.3. Study Selection

In this updated systematic review, studies were eligible if they included the following criteria: (1) they were observational studies that described the prevalence of obesity and excess weight (overweight plus obesity) according to the definitions of the International Obesity Task Force (IOTF) [26,27]; (2) they measured height and weight by trained technicians; and (3) they involved children aged 2 to 13 years. Studies were excluded from the systematic review when: (1) the target population was not a (nationally or regionally or municipality-level) representative sample of the general population of children; (2) studies published in languages other than English or Spanish; (3) when several articles described the same population of children, only the article with the most recent estimates of prevalence was retained; or (4) when authors decided to exclude participants from schools with a low participation rate, we excluded these studies because the published values could underestimate the real prevalence of overweight or obesity [28]. 

### 2.4. Data Extraction

Table 1 shows the main characteristics of the selected studies. Information is as follows: (1) First author and study name; (2) Year/s of measurement; (3) Level of representativeness; (4) Population characteristics (age and sample); and (5) Prevalence (overweight, obesity, overweight + obesity). Data extraction was performed independently by two authors (D.M.B.S. and J.P.R.L.) and disagreements were resolved by a third author (AS). When longitudinal studies published prevalence values only at baseline and final, only the later point measurement was included, for two reasons: to describe the most updated prevalence data and also to avoid studies with longitudinal designs having more influence in the overall synthesis than cross-sectional studies.

### 2.5. Statistical Analyses

Prevalence estimates were pooled using the Metaprop command with Stata SE software, version 16.1 (StataCorp., College Station, TX, USA). The overall prevalence estimates were weighted (in the whole sample and in each sex) according to the sample size and the number of children with excess weight (overweight plus obesity) or obesity in each study. Heterogeneity of prevalence values across studies was evaluated with the *I*^2^ statistic [29]. The DerSimonian and Laird random-effects method was used to calculate the pooled prevalence estimate when substantial or large heterogeneity existed among studies (50–100%) [30]. Mantel–Haenszel fixed-effects method was used if I^2^ was lower than 50%. Confidence intervals for the pooled estimates were obtained using Freeman–Tukey double arcsine transformation (using ftt command in Metaprop, New York, NY, USA) [31]. To describe the trends of obesity and excess body weight (overweight plus obesity), pooled estimates were grouped into two periods (BMI measurements from 1999 until 2010; and from 2011 until 2021).

## 3. Results

### 3.1. Study Selection and Characteristics

Appendix A shows the PRISMA 2020 flow diagram for the studies included in our updated systematic review of the prevalence of childhood obesity and excess body weight (overweight plus obesity) in Spain between 1999 and 2021. From the list of studies included in a previous systematic review [4], nine studies met our eligibility criteria [12,13,15,16,17,19,20,21,22]. In the updated systematic review, a total of 1252 articles were screened between January of 2018 and December of 2021, of which 4 articles met the inclusion criteria [32,33,34,35]. Table 1 contains the 13 articles included in the systematic review. Five studies were representative at the national level [12,21,22,34,35] and eight at the national level [13,15,16,17,19,20,32,33]. 

**Table 1 ijerph-19-05240-t001:** Characteristics of studies (*n* = 13) included in the systematic review of prevalence of overweight, obesity and excess body weight (overweight plus obesity) in Spanish children from 1999 to 2021.

First Author/Study Name	Year	Level of Representativeness	Population Characteristics		Prevalence		
			Age (Years)	Sample (*n*)	Overweight	Obesity	EBW
Serra-Majem J et al., 2006 [12] enKid study	1990–2000	National	2–5 6–13	Total: 385 Boys: 195 Girls: 190 Total: 990 Boys: 492 Girls: 498	16.6% 13.6% 19.9% 18.5 % 26.3 % 20.8 %	10.4% 8.3% 12.7% 7.1 % 8.7 % 5.2 %	27.0% 21.9% 32.6% 30.4 % 35.0 % 26.0 %
Martínez-Vizcaíno V et al., 2008 [13] The CUENCA study	2004	Regional (Cuenca)	9–10	Total: 1166 Boys: 557 Girls: 609	22.0% 21.9% 22.1%	8.8% 10.1% 7.6%	30.8% 32.0% 29.7%
Ara I et al., 2007 [15] Children from Aragón	2000	Regional (Aragón)	7–12	Total: 1068 Boys: 558 Girls: 510	25.0% 25.0% 25.0%	6.4% 5.0% 8.0%	31.4% 30.0% 33.0%
Ahrens W et al., 2014 IDEFICS [16]	2007–2008	Regional (Huesca and Zaragoza)	3–6.99 7–9.99	Total: 928 Total: 509	12.3% 19.8%	5.4% 7.1%	17.7% 26.9%
García-García E et al., 2013 [17] Children from Almería	2007–2010	Regional (Almería)	2–6 6–12	Total: 1068 Total: 504	13.6% 31.0%	8.0% 11.6%	21.6% 42.6%
Brug J et al., 2012 [19] The ENERGY project	2010	Regional (Aragón)	10–12	Total: 1008 Boys: 485 Girls: 523	21.7% 22.9% 20.7%	3.0% 2.9% 3.1%	24.7% 25.8% 23.8%
Gulías-González R et al., 2014 [20] Adolescents from Castilla-La Mancha	2010	Regional (Castilla-La Mancha)	6–11	Total: 1509 Boys: 901 Girls: 917	26.9% 26.7% 27.1%	11.0% 11.9% 10.0%	37.9% 38.6% 37.1%
Wijnhoven T et al., 2014 [21] WHO European Childhood Obesity Surveillance Initiative	2009–2010	National	6 7–9	Total: 1818 Boys: 901 Girls: 917 Total: 5838 Boys: 2938 Girls: 2900	18.5% 16.9% 20.1% 24.6% 23.6% 25.7%	9.3% 8.7% 9.9% 9.8% 9.8% 9.8%	27.8% 25.6% 30.0% 34.4% 33.4% 35.5%
García-Solano M et al., 2021 [35] ALADINO study 2019	2019	National	6–9	Total: 16,665 Boys: 8513 Girls: 8152	22.0% 20.5% 23.6%	10.9% 10.5% 11.4%	32.9% 31.0% 35.0%
Sánchez-Cruz JJ et al., 2013 [22]	2012	National	8–13	Total: 648	25.3%	9.6%	34.9%
Sánchez-Cruz JJ et al., 2018 [32] Andalusian Health Survey	2015–2016	Regional (Andalucía)	8–13	Total: 709	26.2%	13.7%	40.3%
Perez-Rios M et al., 2018 [33] Children from Galicia	2013–2014	Regional (Galicia)	6–11	Total: 4434	25.6%	9.3%	34.9%
Aranceta-Bartrina J et al., 2020 [34] The EMPE study	2014–2015	National	3–8	Total: 396 Boys: 217 Girls: 179	23.6% 23.4% 23.9%	16.3% 14.8% 17.8%	39.9% 38.2% 41.7%

EBW: Excess body weight (overweight plus obesity).

The analyses involved a total of 34,813 children aged 2 to 13 years, with sample sizes between studies averaging 2678 (range: 396 to 16,665). The 13 included studies involved data acquired between 1999 and 2019, with 8 studies from the period 1999–2010 and 5 studies from the period 2011–2021.

### 3.2. Time Trends in the Prevalence of Obesity among Children in Spain

Table 1 shows the prevalence of obesity among children aged 2 to 13 years in Spain from 1999 until 2021 (using IOTF criteria). From 1999 to 2010, the highest prevalence of obesity was 12.7% in girls (2–5 years) [12], and from 2011 to 2021, it was 17.8% in girls (3–8 years) [34]. Eight studies reported separately the prevalence of obesity by sex. The prevalence of obesity was higher in girls than boys in six studies [12,15,19,21,34,35] and in the latter study, only for girls in the 2–5-years-old age group [12] and in Wijnhoven et al. [21] in 6- year-old girls.

### 3.3. Time Trends in the Prevalence of Excess Body Weight (Overweight plus Obesity) among Children in Spain

Table 1 shows the prevalence of excess body weight (overweight and obesity) among children aged 2 to 13 years in Spain from 1999 until 2021 (using IOTF criteria). From 1999 to 2010, the highest prevalence of excess body weight (overweight plus obesity) was 42.6% in both sexes (aged 6–12 years) [17] and from 2011 to 2021, it was 41.7% in girls (3–8 years of age) [34].

Eight studies reported separately the prevalence of excess body weight (overweight plus obesity) by sex. The prevalence of excess body weight (overweight plus obesity) was higher in girls than boys in five studies [12,15,21,34,35]. However, this was only the case for girls aged between 2 and 5 years in one study [12]. In contrast, the prevalence of excess body weight (overweight plus obesity) was higher in boys than girls in four studies [12,13,19,20]. However, this was only the case for boys aged between 6 and 13 years in one study [12]. In contrast, in the prevalence of excess body weight (overweight plus obesity) was higher in boys (than girls) in three studies [12,13,20], yet in the latter study, only for the age group of 6 to 13 years old [12]. In one study [21], the prevalence of obesity was identical in (7–9-year-old) boys and girls. 

### 3.4. Changes in the Prevalence of Excess Body Weight (Overweight plus Obesity) and Obesity (1999–2010 and 2011–2021)

Table 2 displays trends in the pooled prevalence estimates and trends of excess body weight (overweight plus obesity) in children aged 2 to 6 years and in children aged 7 to 13 years between the periods 1999–2010 and 2011–2021. In both sexes, the prevalence of excess body weight (overweight plus obesity) in Spanish children aged 2 to 6 years increased from 23.3% (95% CI, 18.5% to 25.5%) during the period 1999–2010 to 39.9% (95% CI, 35.4% to 44.7%) during the period 2011–2021. In this age-group, the prevalence of obesity (overweight plus obesity) also increased from 8.1% (95% CI, 6.1% to 10.2%) during 1999–2010 to 16.3% (95% CI, 13.0% to 20.0%) during 2011–2021. 

In children aged 7 to 13 years, the prevalence of excess body weight (overweight plus obesity) increased from 32.3% (95% CI, 29.1% to 35.6%) during the period 1999–2010 to 35.3% (95% CI, 32.9% to 37.7%) during the period 2011–2021. In children aged 7 to 13 years, the prevalence of obesity also increased, from 7.8% (95% CI, 6.0% to 9.9%) during 1999–2010 to 10.6% (95% CI, 9.3% to 12.1%) during 2011–2021. 

### 3.5. Changes in the Prevalence of Excess Body Weight (Overweight plus Obesity) and Obesity (1999–2010 and 2011–2021) by Sex

Table 1 includes 13 studies that showed the prevalences of either obesity or excess body weight (overweight plus obesity) in the total sample (both sexes) of children aged 2 to 13 years. However, only eight studies reported additional prevalence values by sex (Table 2). In boys aged 2 to 6 years, the prevalence of either excess body weight (overweight plus obesity) or obesity increased from the period 1999–2010 to the period 2011–2021, by 13.3 and 6.3 percentage points, respectively. In girls aged 2 to 6 years, the prevalence also increased (by 11.3 and 7.5 percentage points for excess body weight (overweight plus obesity) and obesity, respectively). Finally, in boys aged 7 to 13 years, the prevalence of excess body weight (overweight plus obesity) slightly decreased from 32.5% (95% CI, 29.4% to 35.7%) during the period 1999–2010 to 31.0% (95% CI, 30.0% to 32.0%) during the period 2011–2021. However, the prevalence of obesity increased from 7.8% (95% CI, 5.4% to 10.6%) to 10.5% (95% CI, 9.5% to 11.5%) during the same periods. In girls aged 7 to 13 years, the prevalence of excess body weight (overweight plus obesity) increased from 30.9% (95% CI, 26.8% to 35.1%) to 35.0% (95% CI, 34.0% to 36.0%), and for obesity, the value increased from 7.2% (95% CI, 5.1% to 9.5%) to 11.4% (95% CI, 10.4% to 12.4%).

## 4. Discussion

This systematic review and meta-analysis provides the most up-to-date prevalence data of excess body weight (overweight plus obesity) and obesity among Spanish children (2 to 13 years) from 1999 until 2021. Our results indicate that the childhood obesity epidemic in Spain has not plateaued. Further, our findings show dramatic increases in the prevalence of both excess body weight (overweight plus obesity) and obesity during the second decade of this millennia (compared with the first decade), particularly in younger children. Current trends in overweight and obesity are a “ticking time bomb” [36] for many nations due to the considerable negative economic and health impacts caused by the obesity epidemic. Among high-income countries, it is thought that two primary factors contribute to the obesity epidemic: reduced physical activity–related energy expenditure due to environmental or lifestyle-related changes and the increased availability and accessibility of calorically dense, ultra-processed foods and beverages [36]. To reverse the negative trends in obesity and non-communicable diseases in Spain, five priority policies were recently proposed by public health nutritionists to create healthier food environments [37]: (1) regulation of unhealthy food and beverage advertisements targeting children; (2) promotion of “healthy” foods in vending machines located in educational, health and community sport settings; (3) limiting the accessibility to ultra-processed food and beverages through the creation of taxes, accompanied by subsidies or reduced taxes on healthy foods; (4) labeling of foods and beverages with health ratings (Nutriscore); and (5) reformulation of unhealthy products. While this study provides the most up-to-date findings relating to the prevalence of childhood obesity in Spain, the results of the analyses are in contrast to a previous study by Garrido-Miguel et al. [4]. This notable divergence may have been influenced by the inclusion of more recent studies of prevalence (for example, the ALADINO survey (2019), or EMPE study); however, Garrido-Miguel et al. [4] included and pooled results from studies of non-representative samples of Spanish children to describe the prevalence of overweight and obesity (Appendix A), which may have further contributed to the contrasting results. Recruiting a representative (probability) sample is an important prerequisite to make inferences about the general population [38]. Of note, the prevalence of obesity in two non-representative studies [23,24] was notably lower than the overall estimate reported in their meta-analysis, which may have artificially stabilized the trends in obesity and overweight prevalence among Spanish children. On the other hand, Garrido-Miguel et al. [4] pooled the prevalence data from Aguilar Cordero et al. [11], which comprised 325 children of both sexes, aged 9–12 years. However, in the latter publication, the authors did not report the number of participants who were boys or girls. Therefore, it is unclear how Garrido-Miguel et al. [4] could compute the joint prevalence for both sexes from the study by Aguilar Cordero et al. [11], and according to their study protocol, this data should have been excluded [7]. 

The present study has limitations which should be taken into account when interpreting the findings. First, only studies that reported the prevalence of obesity or overweight using IOTF criteria were included. Prevalence values of adiposity varies with the definition of obesity or overweight used, so we excluded prevalence studies that did not follow this criterion. Second, we found a lower number of prevalence studies during 2011–2021 (*n* = 5) than during 1999–2010 (*n* = 8). As a result, further studies would be required to determine more accurate results. Third, there was a relatively low number of eligible studies (*n* = 13). Despite finding large statistical heterogeneity in all meta-analyses, we did not explore the sources of heterogeneity (through subgroup meta-analyses or meta-regression analyses) due to the scarce number of studies for each analysis [30]. Similarly, study quality was not evaluated due to the reduced number of prevalence studies found in Spain. Ideally, sensitivity analyses should have been performed according to the quality scores found in the prevalence studies.

## 5. Conclusions

The prevalence of childhood obesity and excess body weight (overweight plus obesity) in Spain remains very high and has substantially increased (especially in younger children aged 2–6 years) during the past decade. As a result, it can be inferred that the public health measures employed to address the obesity epidemic in Spain have not produced the expected effects, suggesting that new food policies may be required to reverse the excess body weight observed in Spanish children. 

## Figures and Tables

**Table 2 ijerph-19-05240-t002:** Trends in the prevalence (pool estimate (95% confidence interval) of excess body weight (overweight plus obesity) and obesity in Spanish children aged 2–6 years and 7–13 years, using IOTF definition criteria.

	1999–2010				2011–2021				Δ Excess Body Weight 1999–2021		Δ Obesity 1999–2021	
	2–6		7–13		2–6		7–13		2–6	7–13	2–6	7–13
	EBW	Obes	EBW	Obes	EBW	Obes	EBW	Obes				
**Total**	23.3 (18.5 to 25.5)	8.1 (6.1 to 10.2)	32.3 (29.1 to 35.6)	7.8 (6.0 to 9.9)	39.9 (35.4 to 44.7)	16.3 (13.0 to 20.0)	35.3 (32.9 to 37.7)	10.6 (9.3 to 12.1)	+16.6	+3	+8.5	+2.8
**Boys**	24.9 (22.3 to 27.5)	8.5 (6.9 to 10.3)	32.5 (29.4 to 35.7)	7.8 (5.4 to 10.6)	38.2 (31.8 to 44.6)	14.8 (10.5 to 19.8)	31.0 (30.0 to 32.0)	10.5 (9.5 to 11.5)	+13.3	−1.5	+6.3	+2.7
**Girls**	30.4 (27.7 to 33.2)	10.3 (8.6 to 12.2)	30.9 (26.8 to 35.1)	7.2 (5.1 to 9.5)	41.7 (35.0 to 48.3)	17.8 (13.1 to 23.5)	35.0 (34.0 to 36.0)	11.4 (10.4 to 12.4)	+11.3	+4.1	+7.5	+4.2

EBW: Excess body weight (overweight plus obesity); Obes: Obesity.

## Data Availability

All data generated or analyzed during this study are included in this article. Further enquiries can be directed to the corresponding author.

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
