# Peer review of "Has the Prevalence of Childhood Obesity in Spain Plateaued? A Systematic Review and Meta-Analysis"

_ijerph, 2022, doi:10.3390/ijerph19095240_

Round 1

Reviewer 1 Report

The paper is well done and well described; the results are interesting and well described. However, the manuscript is written in poor English and therefore it must be edited:

Line 2 plauteaued – corrected plateaued?

Line 14: prevalence excess – correct: prevalence of excess

Line 60: recruited in each sex – correct: recruited of each sex

Line 65: is key – correct: is a key

Line 97: with to a low – correct: with a low

Line 114: obesity of each study – correct: obesity in each study

Line 138: Table 1 – somewhere is mentioned Serra-Majem J et al. 2006 (with initial) and elsewhere without initial e.g. Martínez-Vizcaíno et al. 2008 - similarly below – please correct

- in line García-García et al. 2013, similarly in the line by Sánchez-Cruz JJ et al. 2018 – please centre the data

Line 158: in the prevalence – correct: the prevalence

Line 159: in three studies 13,20,12 - elsewhere it is stated in parentheses e.g. line 154: in five studies [15,21,34,35,12] – please correct.

Line 160: [12].In – correct [12]. In (space between words)

Line 162: excess of body weight and obesity – in line 149 is stated: excess of body weight (overweight and obesity) – please include the terms correctly and elsewhere in the text too

Line 165: double space between words

Line 215: may have influenced – correct: may have been influenced

Line 224: childhood – correct: children

The discussion is poorly written and must be written more concisely.

The conclusion of the paper is poor. It is not clear what new knowledge the work brings.

Please go thoroughly through the guide for authors and made necessary technical corrections to your manuscript and please correct the grammar – the same is marked directly in the text.

Conclusion: The paper should be accepted for publication with minor revisions.

Although the study has several limitations, it points to a serious problem that needs to be addressed urgently and responsibly.

Reviewer 2 Report

Although the paper analyzes for a long time the evolution of obesity being a review article does not bring relevant data. The percentage increase in obesity is a fact already known in all countries and analyzed by national and European statistical institutes.
I believe that the paper does not bring new information in the research of overweight and obesity

Reviewer 3 Report

Please explain whether the same article can be written without confronting the work listed in item 4. In my opinion, the content presented in this way, being in conflict with other data, constitutes a completely unnecessary polemic, which draws attention more than the content of the work itself.
The work is interesting and worth publishing, but in a revised form. Below is my suggestion to improve the whole.
1) the goal should not refer to other works and criticize them, similarly to conclusions, but refer to the data collected by the authors and what they want to convey to the recipients
2) discussion can and should be a polemic with other data / results
3) conclusions should relate to the topic, purpose and be consistent in the abstract and at the end of the paper
This approach is safer and less conflicting, the more so as the presented work also has its limitations, as the authors write about.

Reviewer 4 Report

The article titled „Has the prevalence of childhood obesity in Spain plauteaued? 2 A systematic review and meta-analysis” has been presented for my review.

Kindly find below few of my comments regarding the manuscript.The introduction is too generic in my opinion. There is missing information on how childhood obesity in Spain correlates with obesity, e.g. in other European countries

[152], [169]- the publications cited in the work, do not present data for 2020-2021, as the latest Garcia-Solano publication is based on data for 2019. Please provide required explanation or change chapters’ titles and tables descriptions. It seems to me that it would be advisable to limit timeframes to 2019, because years 2020-2021 were highly affected by the Covid-19 epidemic and apparent radical changes in lifestyle, eating habits and forms of physical activity, which are certainly parameters disturbing the obesity scale. In my opinion, it is necessary to bring up this topic at the point of discussion. The manuscript appears to be written as a report I think it would be beneficial to expand the discussion as to why the situation in Spain regarding child obesity is this way. Are there any countries which handled the problem the better? Are there proven models, which could be utilized in Spain? Although the work is a reviewer's manuscript, the author's own conclusions should be included. For above reason, the quotation of [39] in the conclusions is inappropriate.

Round 2

Reviewer 2 Report

Accepted

Reviewer 3 Report

Thank the authors for introducing the changes I suggest. The new version is much more pleasant to read, and the conclusions are based only on the review of the research included in this work. It's a good job.